# Shifting from a Biological-Agnostic Approach to a Molecular-Driven Strategy in Rare Cancers: Ewing Sarcoma Archetype

**DOI:** 10.3390/biomedicines11030874

**Published:** 2023-03-13

**Authors:** Aldo Caltavituro, Roberto Buonaiuto, Erica Pietroluongo, Rocco Morra, Fabio Salomone, Pietro De Placido, Martina Pagliuca, Angelo Vaia, Margaret Ottaviano, Marianna Tortora, Sabino De Placido, Giovannella Palmieri, Mario Giuliano

**Affiliations:** 1Department of Clinical Medicine and Surgery, University of Naples Federico II, 80131 Naples, Italy; 2Scuola Superiore Meridionale (SSM), Clinical and Translational Oncology, 80138 Naples, Italy; 3Gustave Roussy, 94805 Villejuif, France; 4Medical Oncology Unit, “San Carlo” Hospital, 85100 Napoli, Italy; 5Unit of Melanoma, Cancer Immunotherapy and Development Therapeutics, Istituto Nazionale Tumori IRCCS Fondazione Pascale, Campania, 80131 Napoli, Italy; 6CRCTR Coordinating Rare Tumors Reference Center of Campania Region, 80131 Naples, Italy

**Keywords:** ewing sarcoma, rare thoracic tumors, epigenetics, immunotherapy, biology

## Abstract

Sarcomas of the thoracic cavity are rare entities that predominantly affect children and young adults. They can be very heterogeneous encompassing several different histological entities. Ewing Sarcoma (ES) can potentially arise from every bone, soft tissue, or visceral site in the body. However, it represents an extremely rare finding when it affects the thoracic cavity. It represents the second most frequent type of thoracic sarcoma, after chondrosarcoma. ES arises more frequently in sites that differ from the thoracic cavity, but it displays the same biological features and behavior of extra-thoracic ones. Current management of ES often requires a multidisciplinary treatment approach including surgery, radiotherapy, and systemic therapy, as it can guarantee local and distant disease control, at least transiently, although the long-term outcome remains poor. Unfortunately, due to the paucity of clinical trials purposely designed for this rare malignancy, there are no optimal strategies that can be used for disease recurrence. As a result of its complex biological features, ES might be suitable for emerging biology-based therapeutic strategies. However, a deeper understanding of the molecular mechanisms driving tumor growth and treatment resistance, including those related to oncogenic pathways, epigenetic landscape, and immune microenvironment, is necessary in order to develop new valid therapeutic opportunities. Here, we provide an overview of the most recent therapeutic advances for ES in both the preclinical and clinical settings. We performed a review of the current available literature and of the ongoing clinical trials focusing on new treatment strategies, after failure of conventional multimodal treatments.

## 1. Introduction

Ewing’s sarcoma (ES) is an extremely rare malignancy with an incidence of one case per million and a prevalence of 200 cases in the United States [1]. Overall, ES accounts for less than 5% of all sarcomas, although between young adults and adolescents it can reach 10% to 15% of all bone sarcomas [1].

ES of the chest wall is also known as an Askin tumor and shows the same genetic lineage of ES of the bone [2].

Although the diagnosis of ES can be supposed by immunohistochemical (IHC) analysis, the molecular confirmation with the detection of the fusion transcript EWRS1-FLI1 is nowadays mandatory to achieve a proper histological characterization and to be able to differentiate ES from other round-cell sarcomas (RCSs) on biopsy. Clusters of differentiation 99 (CD99) and the Friend leukemia integration 1 transcription factor (FLI-1) are currently accepted for the initial IHC diagnosis of ES, but they can also be expressed in other cancers [3].

ES is characterized by several different genetic rearrangements, but the most frequent involves the 11th and 22nd chromosomes in t(11;22) (q24;q12), representing a non-random rearrangement between the Ewing sarcoma (EWS) gene and the E26 transformation-specific or E-twenty-six (ETS) gene family, which generates the fusion gene EWSR1-ETS [4]. This translocation is reciprocal and involves the Ewing sarcoma breakpoint region 1 (EWSR1) gene and FLI1, a gene belonging to the ETS family. EWSR1 and FLI1 merge and create a fusion gene that codifies for a chimeric protein. This fusion accounts for about 90% of ES cases while the remaining 10% is represented by the fusion of EWSR1 with the ETS-related gene (ERG) in the translocation t (21;22) (q11;q12) [5]. FLI1 and ERG are both members of the ETS transcription factor family, which shares a conserved DNA binding domain structure.

Nowadays, ES treatment mostly requires a multidisciplinary approach which is highly focused on the patient’s personal features, such as age, comorbidities, and single preferences, as well as on tumor features, including site, size, and local extension.

Although surgery, radiotherapy, and systemic treatment, can provide considerable survival benefit in presence of localized disease, patients with advanced stage ES still have a poor outcome [6].

Several chemotherapy (CT) agents, including vincristine, doxorubicin, cyclophosphamide, ifosfamide, and etoposide have demonstrated activity against ES in large collaborative trials. These compounds are frequently combined in regimens that are used both as induction and consolidation systemic therapies [7,8,9]. In localized disease, an up-front surgical approach, when feasible, should always be taken into considered. A neoadjuvant systemic treatment may be required to guarantee the optimal surgical resection, if initially not achievable. Unfortunately, ES eventually recurs, and efforts have been made to detect, among the regimens currently available, the one or more regimens capable to significantly impact progression free survival (PFS).

The randomized phase III rEECur trial provided the first comparison among different CT regimens used in recurrent or primary refractory ES. The four most common CT schemes including irinotecan plus temozolomide (IT), gemcitabine plus docetaxel (GD), high dose ifosfamide (hd-IFO), and topotecan plus cyclophosphamide (TC), were compared among each other. After the first and second interim analysis the two least effective treatment arms, namely GD and IT, were discontinued.

In the final analysis, hd-IFO showed significantly superior PFS and overall survival (OS), as compared to TC [10]. Currently, the rEECur trial is recruiting patients treated with hd-IFO or an additional, recently added arm, carboplatin plus etoposide (CE). Data are still awaited, as a molecularly targeted agent arm is planned [10]. In addition, several ongoing clinical trials are investigating new potential treatment strategies, based on ES biological and molecular features.

ES retains unique biological characteristics that could be translated into therapeutic opportunities after the failure of standard treatments. A comprehensive evaluation of the molecular mechanisms driving tumor growth and treatment resistance, including those related to oncogenic pathways, epigenetic landscape, and immune microenvironment, could allow to create a paradigmatic treatment shift in ES.

Here we present an overview of the present and future therapeutic perspectives for relapsed and refractory ES, based on the molecular pathways active in this rare cancer.

## 2. Receptor Tyrosine Kinases (RTKs)

Receptor tyrosine kinases (RTKs) are transmembrane receptors involved in cell proliferation and survival.

In ES, the activation and overexpression of RTKs have been widely observed suggesting their crucial role in the pathogenesis of this disease and in paving the way for the development of drugs directed against these receptors.

The most common RTKs involved in ES biology are insulin-like growth factor 1 receptor (IGF1R), C-KIT, platelet-derived growth factor receptor β (PDGFR-β), epithelial growth factor receptor (EGFR), and vascular endothelial growth factor receptor 2 (VEGFR-2).

### 2.1. Insulin-like Growth Factor 1 Receptor (IGF1R) in ES

IGF1R is a transmembrane receptor composed by two subunits, ⲁ and β, connected by disulphide bonds. The first component, the ⲁ subunit, is responsible for ligand binding whereas the other, the β subunit, upon autophosphorylation leads to the activation of several intracellular pathways involved in cell proliferation and survival such as the mitogen-activated protein kinase (MAPK or MAP kinase) pathway and the phosphatidylinositol-3-kinase (PI3K)/Akt and the mammalian target of rapamycin (mTOR) pathway. Both pathways drive oncogenic transformation, proliferation, and survival [11].

The relation between the fusion protein EWSR1-ETS and IGF1R pathway is complex and not completely understood.

Preclinical evidence has shown that IGF1R is ubiquitously expressed in neuroectodermal-originating tumors and its activation is sustained by the autocrine production of insulin-like growth factor-1 (IGF-1), one of the IGF1R ligands, by the cancerous cells themselves [12].

Furthermore, EWSR1-ETS acts both as a transcription factor of the IGF-1 and IGF-2 encoding genes and as a negative regulator of the sequester of the circulating IGF-1, the insulin-like growth factor binding protein 3 (IGFBP3). In fact, EWSR1-ETS once binding IGFBP3 promoter reduces its expression and as a consequence IGF-1 can bind IGF1R [13,14,15].

The prognostic role of IGF1R expression has been proved in clinical samples of ES. Indeed, a better prognosis and a lower tumor proliferative rate when IGF1R is less expressed has been demonstrated [16].

The existence of this crosstalk has paved the way to the investigation, as has the development, of antibodies and small molecules, both in preclinical and clinical stages, that could interfere with this biological interaction (Figure 1A).

Monoclonal antibodies directed against IGF1R

R1507, a monoclonal antibody (mAb) directed against IGF1R, has proven to be active in the growth inhibition of ES cells that expressed high levels of IGF-2 and has shown encouraging activity and a durable benefit in a phase II trial, with acceptable toxicities. In this trial the OS was 6, 9 months with an overall response rate (ORR) of 10% and a duration of response (DOR) of 28 weeks [17].

IGD1R inhibitors have gained momentum and apart from R1507, other antibodies have demonstrated activity against IGF1R: dalotuzumab, known as MK-0646, also showed in vitro and in vivo activity against ES cells, as well as IMC-A12 (cixutumumab), AMG 479 (ganitumab), CP-751, 871 (figitumumab), and SCH-717454 [18,19].

In a phase II trial ganitumab has demonstrated a median PFS of 7.9 months in patients with metastatic ES [20].

In the light of the potential synergism between IGF1R inhibitors (IGF1Ri) and conventional chemotherapy, the addition of ganitumab to chemotherapy in newly diagnosed metastatic ES patients was investigated in a phase III trial [21].

Unfortunately, no event-free survival (EFS) benefit with the addition of the anti-IGFR1 mAb to interval-compressed chemotherapy, consisting of vincristine/doxorubicin/cyclophosphamide alternating once every two weeks with ifosfamide/etoposide (CDV/IE), was seen [21].

Increased toxicity, mainly febrile neutropenia and the elevation of alanine aminotransferase, was associated with ganitumab.

Moreover, figitumumab, another mAb directed against IGF1R, has shown preliminary encouraging results in a phase II trial in patients with ES, particularly when elevated blood IGF-1 levels were detected at the baseline. As a matter of fact, a longer OS was observed in subjects with baseline IGF-1 levels > 110 ng/mL compared to patients with lower IGF-1 levels at the baseline. However, the mPFS in the overall population was very poor reaching only 1.9 months [22,23]. This finding suggests that a better selection of patients should be pursued to observe better results. As an example, preliminary analysis of IGF-1 levels may represent a strategy to understand which patients would benefit the most from IGF1R inhibition.

The modest activity of these drugs, emerging from the above-described trials, could probably be due to the heterogeneity of IGF1R distribution on the cells surface and to the overexpression of IGFBP3 which represent a secondary resistance mechanism.

In order to increase the clinical activity of IGF1Ri, their association with other drugs, such as mTOR inhibitors, have been investigated. In fact, pre-clinical data suggest that a deletion of the phosphatase and tensin homolog on chromosome 10 (PTEN), involved in the mTOR pathway, can be identified in 25% of patients with ES [24]. This genetic alteration confers resistance to IGF1Ri explaining the rationale of combining these two drugs.

Furthermore, the IGF1R overexpression represents a resistance mechanism to different conventional and not conventional drugs.

As an example, IGF1R is a mediator of resistance to target drugs, such as cyclin dependent kinase 4/6 inhibitors (CDK4/6i).

The CDK4/6 pathway is active in several cancers and shows a dependency in ES on the IGF1R pathway.

In the cell cycle regulation both CDK4 and CDK6 mediate the transition from the G1 phase to the S phase in order to prime DNA replication. Upon activation by the CDK-activating kinase (CAK) complex, both CDK4 and CDK6 inhibit, by phosphorylation, retinoblastoma (RB), a tumor suppressor protein. RB exerts an inhibitory function on the E2F family of transcription factors whose target genes are implicated in cell proliferation [25].

The consequences of CDK4/6i are the blockade of cell proliferation caused by an arrest in the G1 phase.

Ganitumab, and other IGF1Ri, have shown to be synergistic with CDK4/6i in vitro and active in ES mouse models. For this reason, ganitumab is under investigation in combination with palbociclib, a CDK4/6i. The aim of the trial is to verify if this combination is safe and active against ES [26].

Furthermore, it was demonstrated that IGF1R signaling could protect cancerous cells from the cytotoxic effect of several chemotherapeutic agents active in sarcomas.

Indeed, the efficacy of vincristine, trabectedin, and doxorubicin was proven to increase when IGF1R is inhibited [27,28,29].

Tyrosine kinase inhibitors against IGF1R

Apart from mAbs, IGF1R can also be inhibited by small molecules, namely inhibitors of the intracellular tyrosine kinase domain of IGF1R.

One established resistance mechanism to IGF1R mAbs is represented by the compensatory activation of other RTKs as described below.

Linsitinab, a dual inhibitor of both IGF1R and insulin receptor (IR), whose activation is recognized as one resistance mechanism to anti-IGF1R drugs, has demonstrated to decrease the phosphorylation of the IGF1R thus inducing a reduction of ES cell proliferation.

Its good pharmacokinetic profile made this drug well tolerated [30].

Besides Linsitinib (OSI-906), ADW742 and NVP-AEW54, alone or in combination with conventional CT, have been demonstrated to inhibit cell growth in in vitro and in vivo studies [30,31,32].

In particular, NVP-AEW541 has demonstrated to selectively inhibit IGF1R by distinguishing between IGF1R and other RTKs leading to a more sustainable toxicity profile. Its effects on tumor growth were noticed when NVP-AEW541 was combined with vincristine, actinomycin D, and ifosfamide [32].

It is worth remembering that toxicity profiles of these agents must be considered.

Although IGF-1R mAbs and small inhibitors are well tolerated, particular attention should be paid to mild and not mild adverse effects.

Hyperglycemia was commonly described in all the pre-mentioned trials, probably because IGF-1R is involved in glucose metabolism. Two mechanisms could explain this expected toxicity: compensatory insulin secretion and, consequently, insulin resistance [33].

To summarize, although large amounts of preclinical and clinical data support the use of agents targeting the IGF1R pathway, the real benefit of these compounds is questionable and seems not to apply to every patient. This data suggests that it is possible that combined therapies may represent a suitable strategy to enhance the, at least for now, disappointing anti-tumor activity of anti-IGF1R agents in monotherapy. Predictive biomarkers of the response to IGF1Ri are not available but there is a need to better select patients that may benefit from these drugs.

### 2.2. Other Targetable RTKs in ES

Imatinib, a C-KIT inhibitor, has been tested in patients with ES showing different and contrasting results in some trials even if it seems to sensitize cancerous cells to conventional CT.

The rationale for the use of imatinib [34], or other agents such as olaratumab [35], relies on their inhibition of the KIT/stem cell factor (SCF) receptor and PDGFR-β that are both expressed in ES [36]. A phase II trial has been conducted to evaluate the efficacy of imatinib in seven patients with recurrent ES. Only one of them has shown a modest benefit from imatinib. This patient had the highest levels of PDGFRα and KIT expression. These results, once again, underline the importance of an appropriate selection of patients for target therapy in rare tumors. Unfortunately, only a small percentage of patients affected by ES express high levels of these receptors. Moreover, perhaps, KIT receptors, as well as PDGFR- β, are not considered primary therapeutical targets in this disease [34].

EGFR, whose activation drives proliferation, and angiogenesis are another suitable target in ES and could be inhibited both by small molecules such as gefitinib [37] and antibodies such as cetuximab [38]. Although preclinical trials failed to demonstrate an effect on tumor growth for gefitinib, the EGFR blockade results in an increase in VEGF expression. This provides an alternative, pro-angiogenic pathway that promotes tumor survival which could be targeted with anti-VEGF(R) inhibitors [34].

RTKs inhibitors have also been recently investigated in combination with CT. In a Ib/II phase study the combination of anlotinib, a multi-target tyrosine kinase inhibitor (TKI), with irinotecan and vincristine, has been proved feasible and effective in patients with recurrent or refractory ES. Among the twenty-three patients evaluable, the ORR was 65%. The most common toxicities detected were leukopenia, anemia, and neutropenia, possibly related to the chemotherapy backbone [39].

In conclusion, RTKs represent potential targets in ES that deserve further investigation. Limitations in the development of specific drugs that could have a direct impact on these receptors seems to be due to several factors: heterogeneity in receptor expression, dependency of ES cells on different survival pathways, and existence of escaping and resistance mechanisms.

## 3. EWSR1-FLI1

EWSR1-ETS behaves as a transcription factor and induces the expression of proteins which promote tumorigenesis by switching on and off critical genes. This double role turns this fusion protein into the most prominent genetic alteration that can be targeted in ES [40].

The fusion protein EWSR1-ETS can be targeted at several levels and in different ways. The first approach to decrease a EWSR1-FLI1 related transcription is by inducing an expression impairment. EWSR1-FLI1 activity could be decreased also by targeting its interaction with other proteins involved in its transcriptional functions.

Moreover, EWSR1-FLI1 is able to deregulate some genes whose transcriptional products could be targetable.

### 3.1. Decreasing EWSR1-FLI1 Expression

In order to decrease the expression of the fusion protein both small antisense oligodeoxynucleotides (oligos) and small interfering RNAs (siRNAs) have been investigated in in vitro experiments. The results have shown that the use of these agents against ES cells is safe, and it generates a positive tumor response.

The oligos bind complementary mRNA sequences, in this case the mRNA that codes for EWSR1-FLI1, to primarily stop its translation and subsequently induce its degradation by ribonuclease H [41].

The siRNAs are also able to switch off EWSR1-FLI1 by gene-silencing but the exact mechanism of knockdown is still an area of investigation [42].

It is widely accepted that, upon recognition of targeted sequences, siRNAs mediate a specific and selective cleavage of the target mRNAs interfering with their translation into functional proteins capable of driving oncogenic features such as proliferation, tumor development and progression, and apoptosis evasion [43].

One of these molecules is currently being evaluated in a clinical trial (NCT02736565).

Both oligos and siRNAs interfere with the translation of the fusion gene into the fusion protein and consequently reduce the downstream activity of the same. Although potentially promising agents, both oligos and siRNAs have a major limitation: because of their short half-life, they require repeated administration to achieve and sustain EWSR1-FLI suppression; an expensive and time-consuming procedure [44] (Figure 1B).

### 3.2. Decreasing the Activity of EWSR1-FLI1

As mentioned above, another way to stop the proliferation of ES cells is to decrease the transcription activity of EWSR1-FLI1 by modulating the exons usage directly or indirectly. In fact, EWSR1-FLI1 needs some modulators such as RNA helicase A (RHA) that affect the pre-mRNA processing. This protein is part of the interactome, a complex representation of functional interactions between molecules in ES cells. RHA, with its helicase function, enhances the transcriptional activity of the fusion protein which in return can affect the activity of RHA by inhibiting its helicase function [45].

On these bases, a small molecule, called YK-4-279 has preclinically shown to interrupt the binding of EWSR1-FLI1 to RHA. As a consequence, EWSR1-FLI transcription activity is arrested, and the RHA helicase function is restored. This interference leads to apoptosis in ES cells and reduces growth in ES xenografts. (Figure 1C).

Taking into consideration what was previously described, the disruption of the interactome of ES may represent a suitable approach to treat ES.

It is worth noting that some questions remain unsolved. First and foremost, whether the presence of an ETS translocation is needed for YK-4-279 to be effective or whether an overexpression of an ETS transcription factor, although without a translocation, is enough. Pre-clinical data suggest that a high expression of FLI1 or EWSR1 is not sufficient to induce malignant transformation, and strengthen the hypothesis that EWSR1–FLI1 fusion is a more potent transcriptional activator than FLI1 itself, even if the DNA-binding affinity of EWSR1–FLI1 and FLI1 alone are nearly the same [46].

Secondly, ETS encompasses a wide range of EWS partners, and it remains unclear whether YK-4-279 could be active against all of them [46].

Another small EWSR1-ETS inhibitor, named TK216 is also under investigation.

TK216 is a drug designed in order to inhibit the effect of the EWSR1-ETS transcription factor similarly to YK-4-279. The activity of TK216 is explicated on ETS proteins directly by the disruption of protein—protein interactions, and the inhibition of transcription factor function.

The association of TK216 and vincristine (VCR) exerted synergistic activity in non-clinical models.

A first-in-human study of TK216 in patients with ES was launched in 2016 by JA Ludwig et al. This study was designed to establish the safety profile and to collect efficacy data of this drug either in monotherapy or in combination with VCR. Eventually, the combination of TK216 with a vinca alkaloid was chosen as the most efficient one, as it was underlined that a mutation in TUBA1B, encoding α-tubulin, could drive resistance to TK216 monotherapy (NCT02657005).

In an interim analysis, the clinical benefit rates of TK216+VCR were 46.4%. The authors concluded that TK216 plus VCR was well tolerated and showed encouraging early evidence of anti-tumor activity in a heavily pre-treated, high tumor burden ES population [47].

Pharmacokinetic and pharmacodynamics issues remain unsolved and further studies are needed to predict whether high peak serum concentrations or prolonged lower drug exposures are safe or required in order to achieve maximum antineoplastic activity. Moreover, resistance mechanisms, such as the activation of other proliferation pathways or upregulation of proteins, could be responsible for reduced activity of these drugs.

The development of YK-4-279 and TK216 is still in its infancy, and in conclusion it seems too early to translate them into clinical practice.

## 4. The Epigenomic Landscape of ES

Another hallmark of ES biology is the heterogeneity in its epigenetic profile that may explain, at least in part, the differences between the courses and aggressiveness of the disease in different patients.

The epigenetic profile of ES is linked to the activity of EWSR1-FLI1.

EWSR1-FLI1 expression can induce epigenetic alteration which could transform normal cells into cancerous ones. Epigenomic profiling of ES has shown that EWSR1-FLI1 is responsible for an epigenetic reprogramming of cancerous cells, by inducing de novo enhancers or by repressing enhancers that are normally active [48].

The mechanisms used to alter the epigenetic landscape in ES are several and complex. The most studied one deals with the effects of EWSR1-FLI1 on areas of the genome known as GGAA microsatellite response elements. It might also be noticed that promoters of target genes directly bound by EWSR1-FLI1 exhibit a significant over-representation of these highly repetitive GGAA-containing elements (microsatellites) [49].

Because ES specific enhancers are enriched in GGAA microsatellites, EWSR1-FLI1 can act as a pioneer transcription factor and increase the accessibility to DNA by other factors that modify the chromatin, such as the SWI/SNF complex, a chromatin remodeling set of proteins. The effects that EWSR1-FLI exerts on global transcription derive by the recruitment of acetyltransferases and demethylase on these GGAA areas, with the aim of activating or repressing enhancer elements.

With this biological rationale, the modulation of the epigenome and its components may represent a valid option in treating ES.

Efforts have focused on the use of the epigenome in ES as a therapeutic target with the investigation of several drugs that could modulate and edit it.

### 4.1. Histones Demethylation Inhibitors (HDMi)

EWSR1-FLI1 uses a specific enzyme, named lysine-specific demethylase 1 (LSD1) to repress some critical tumor suppressors, such as lectin-like oxidized low-density lipoprotein (LDL) receptor-1 (LOX1) and transforming growth factor beta receptor 2 (TGFBR2) [50].

LSD1 is known to be overexpressed in ES cancer cells [51] and can be inhibited by HCI2509.

Upon the inhibition of LSD1, a reduction of the expression of the pre-mentioned genes, LOX and TGFBR2, both involved in tumor suppression, occurs. The inhibition of LSD1 leads to an impairment of tissue culture cell viability in multiple ES cell lines as verified by some preclinical trials.

This drug was found to disrupt the transcriptional signature of EWSR1-FLI and EWS-ERG and to silence malignant characteristics of ES cells by reversal of global transcriptional activity of both fusion genes [50] (Figure 2A).

A first report of an LSD1 inhibitor, seclidemstat (SP-2577), in a phase I trial focused exclusively on ES, has shown promising results with a manageable safety profile and a proof-of-concept of preliminary activity in heavily pretreated relapsed or refractory patients.

The most common adverse events related to the treatment were vomiting, abdominal pain, hypokalemia, and hematologic disorders [52].

Seclidemstat deserves more investigation in an already planned phase II expansion trial of seclidemstat, as a single agent and in combination with CT, topotecan and cyclophosphamide, in ES and other sarcomas that share similar translocations (NCT03600649).

Among other molecules that have shown to regulate epigenetic profiling in ES, second-generation histone demethylases (HDM) inhibitors, such as the pan-Jumonji inhibitor JIB-04, have proven activity in preclinical models of ES. This drug targets HDM3, a vector of a significant slowdown in the in vitro and in vivo growth of ES cells [53].

### 4.2. Histones Methyltransferase Inhibitors (HMTi)

An enhancer of Zeste Homologue 2 (EZH2) is involved in a wide range of biological processes, including cell cycle regulation, cell differentiation and proliferation, division, and senescence/apoptosis.

EZH2, the catalytic subunit of Polycomb Repressive Complex 2 (PRC2), is also crucial for maintaining the epigenetic state of the cells, thanks to the modulation of the chromatin structure and, consequently, the regulation of gene expression [54].

From a biological point of view, EZH2 is a histone methyltransferase that can be mutated across different cancers, both hematologic and solid.

A very small percentage of soft tissue sarcomas (STS), and among them ES, can exhibit a mutation in EZH2 or in another component of its belonging complex [55].

Tazemetostat, a EZH2 inhibitor that has shown to be active in epithelioid sarcoma (EpS) [56], has been also investigated in other solid tumors harboring alterations in EZH2 or the SWI/SNF complex.

Unfortunately, the study did not produce significant results, showing an ORR of only 5% [57].

### 4.3. Histones Deacetylase Inhibition (HDACi)

Epigenetic alterations in ES are also a consequence of the activation of histone deacetylases (HDACs) which are responsible for chromatin relaxation and transcriptional activation of genes involved in tumorigenesis.

EWSR1-FLI1 can induce epigenetic alterations through the activation of HDACs that lead normal cells to reprogram themselves into malignant ones.

Two HDACs inhibitors, romidepsin and entinostat, have both shown in vitro and in vivo activity in ES cells [58].

These two drugs can reverse the activation of HDACs and restore the normal epigenetic profile of transformed cells inducing a regression of established ES xenografts, a decrease in the levels of EWSR1-FLI1 mRNA and fusion proteins, inhibition of the DNA synthesis and cell proliferation with an arrest in G1 and G2 phases, and the induction of tumor necrosis factor (TNF)-related apoptosis-inducing ligand (TRAIL)-dependent apoptosis [58].

In the epigenetic editing of DNA, aminoacidic residues such as lysine must be read, once acetylated, by proteins belonging to the family of bromodomain and extra-terminal domain (BET) proteins.

BET proteins act as epigenetic readers recruiting transcription factors and coactivators to target gene sites in order to promote tumorigenesis.

These genome readers can be inhibited by compounds already tested in preclinical settings.

As an example, the BET inhibitor JQ1 seems to reverse the EWSR1-FLI transcriptional signature and, consequently, arrest ES xenografts’ growth [59].

The transcriptional activity of EWSR1-FLI1 can also be directly targeted with drugs belonging to the mithramycin class. Mithramycin is an antibiotic with anti-tumor properties but its toxicity has limited its use in clinical practice so far. Two drugs, derived by mithramycin, although less toxic, EC-8042 and EC-8105, have shown efficacy in ES [60]. Other drugs, such as EnglerinA (EA) have proven activity against ES cells thanks to their ability to directly affect the transcriptional activity of EWSR1-FLI1 by both the dephosphorylation of EWSR1-FLI1, which results in the reduction of the DNA binding capacity of the fusion protein to target genes, and the induction of necrosis and apoptosis [61].

These data strongly suggest that ES is not solely a genetic disease, but it is also characterized by a wide range of epigenetic disorders and knowledge of these is crucial in both defining and treating it.

## 5. DNA Damage Response in ES

Physiologically, a clear role for EWSR1 in normal cells has been established: EWSR1 is crucial in responding to DNA damage, suppressing the formation of R-loops, and promoting homologous recombination (HR) [62].

In ES cells, the aberrant formation of EWSR1-ETS interferes with wild-type EWSR1 in regulating these pathways leading to the historical chemosensitivity and radiosensitivity of this disease [63].

ES is in fact sensitive to ionizing radiation and genotoxic agents, such as doxorubicin, cyclophosphamide, etoposide, and ifosfamide, part of the most effective chemotherapeutic regimens both in the early and advanced setting as previously mentioned.

Thus, this chemosensitivity is increased by the activity of EWSR1-ETS which affects the integrity of the entire genome [64].

### Poly (ADP-Ribose) Polymerase (PARP) Inhibitors (PARPi) in ES

Following EWSR1-ETS translocation and activation, in ES cells there is an aberrant regulation of transcription with an accumulation of R-loops and an increased replication stress which represent endogenous sources of DNA damage. Thus, this malignancy is more sensitive to other drugs targeting one of the candidate genes that mitigates these DNA damages known as poly (ADP-ribose) polymerase (PARP) expressed at high levels in ES cells [65].

Moreover, ES cells display, when compared to normal cells, a significant functional impairment or absence of BRCA1 in the context of EWSR1-FLI1 expression [64].

Knowing that PARP1 inhibition, for example with olaparib or niraparib, exercises a synthetic lethality with a BRCA1 deficiency in replicating cells, an impaired BRCA1 function could provide a molecular basis for the high sensitivity of ES to this class of agents [66].

To better understand what happens in an ES proliferating cell the following scenario is presented: EWSR1-ETS chimeric proteins activate the inhibitor of the DNA binding 2 (Id2) gene either directly or indirectly, as a result of the activation of the c-myc gene. The higher levels of the Id2 protein enhance cell cycle progression by inactivating the RB protein family and releasing E2F, which further promotes PARP-1 expression.

Proliferation stress results in higher levels of DNA damage that requires repair mechanisms. Since BRCA1 is functionally absent in these cells and non-homologous recombination (NHR) cannot be performed; the simultaneous inhibition of PARP1 leads to cell death as not even homologous recombination (HR) can take place (Figure 2B).

Olaparib, a PARP inhibitor (PARPi), was investigated in a prospective trial for patients with refractory ES following failure to standard CT. Unfortunately, laparib did not show a significant or durable efficacy in this populations with no objective responses in the twelve patients enrolled and evaluable [67]. The average time to disease progression (mTTP) was 5.7 weeks and further enrolment was, for these reasons, discontinued.

These data suggest that it is possible that PARPi monotherapy is not sufficient to determine an effective response and a deeper knowledge of biomarkers involved in homologous recombination repair (HRR), such as PALB2 and RAD51 [68,69,70], predictive of response to these therapies, is needed. In order to verify if combination therapy could translate the pre-mentioned biological rationale into tangible clinical benefit, niraparib and talazoparib, other two PARPi, were studied in combination with temozolomide and irinotecan in patients with pre-treated incurable ES. The combination demonstrated tolerability with adjusted doses of CT, but no antitumor activity was observed [68,69].

Beyond PARP, there are other candidate proteins to target in order to inhibit DNA damage response (DRR) in ES cells; another possible regulator of DNA single and double strand breaks (DSSBs-DDSBs) repair is ataxia telangiectasia and Rad3 related protein (ATR) and its pathway.

Two ATR inhibitors, AZ20 and MSC253, alone or in combination with inhibitors of downstream effector of ATR such as WEE1 [71], were investigated in preclinical trials, demonstrating some efficacy [72,73].

## 6. The Downstream Effectors of EWSR1-FLI1

### 6.1. Aurora Kinase (AURK) Inhibitors

Among the genes upregulated by EWSR1-FLI1, three aurora kinase (AURK) family members (A, B, and C) are crucial for tumorigenesis.

Aurora kinases are key regulators of several transduction pathways; their activity is regulated by the mitogen-activated protein kinase/extracellular signal-regulated kinase (MAPK/ERK) pathway [74].

Here are some of the many roles covered by these threonine-serine kinases: the control of the centrosome and kinetochore function in the assembling and activity of the spindle assembly cell checkpoint (SAC), the mitotic entry, and the interaction with critical proteins such as p53 and MYC [74].

The overexpression of AURK has been proved to be important in several solid and hematologic neoplasms [74].

EWSR1-FLI1 directly contacts the promoters of AURK-A/B/C resulting in their overexpression. For this reason, AURKs can be considered suitable targets for the treatment of ES. The discovery of this biological dependency has paved the way for the development of inhibitors such as alisertib, an AURK-A inhibitor, and tozasertib, a pan-AURK inhibitor. They both have shown in vitro and in vivo efficacy to inhibit cell growth [75] (Figure 3).

To date, some trials are investigating the use of AURK inhibitors (AURKi) with conventional CT even if the spectrum of adverse events, prevalently hematological toxicities, prevent these combinations from reaching clinical practice.

Aurora kinases, in particular AURK-A, plays a crucial role also in DDR and modulates the DDSBs’ repair through the inhibition of the recruitment of RAD51, a major effector of the DNA repair system [76].

These results underlined a new function of AURK-A suggesting a potential therapeutic approach for the treatment of cancers overexpressing this protein that consists of combining PARPi and AURKi to enhance the efficacy of the latter [77].

AURKi have shown better results in hematological neoplasms rather than solid ones. One possible explanation may relate to the proliferation rate of cells in solid tumors that is relatively slow compared to the ones of non-solid tumors. Further research is needed to understand whether the simultaneous inhibition of a product of an activated oncogene and AURK may be a suitable strategy to achieve a significantly improved clinical outcome and overcome upcoming resistance.

### 6.2. Forkhead Box O (FOXO)1

Other two transcription factors have shown a potential role in the inhibition of ES cells when specific drugs were used to antagonize their activities.

Consolidated evidence suggests that the EWSR1 domain confers transcription activating properties to the EWSR1-FLI1 chimeric protein; whereas it is also known that the fusion protein has transcription repressing properties. EWSR1-FLI1-repressed genes show enrichment of forkhead box (FOX) recognition motifs.

Forkhead box O (FOXO)1 is a transcription factor involved in cell proliferation, differentiation, and apoptosis; it acts as a tumor suppressor that limits the growth of cancerous cells inducing cell-cycle arrest, apoptosis, and DNA repair [78].

FOXO1 is transcriptionally and post-transcriptionally repressed by EWSR1-FLI1 meaning that the re-activation of FOXO1 may represent a promising strategy for ES. Methylseleninic acid (MSA), a chemical compound that has proved to reactivate FOXO1 in prostate cancer [79], was investigated also in ES. A proof of principle study was undertaken demonstrating a dose- and time-dependent reactivation of endogenous FOXO1 after MSA administration. Furthermore, MSA has proved to induce cell death in a FOXO1 expression-dependent manner with a subsequent reduction in tumor growth in vivo [80].

### 6.3. Glioma-Associated Oncogene Homolog 1 (Gli1)

Moreover, Glioma-Associated Oncogene Homolog 1 (Gli1), a transcription effector of the canonical Hedgehog (HH) pathway, shown to be highly expressed in ES, was known to mediate the downstream effects of the fusion protein. Gli1 is a direct target of EWSR1-FLI1 and seems to mediate the developmental process and the maintenance of the stem-cell-reservoir confirming the aggressiveness of ES cells [81].

When Gli1 levels were reduced by Arsenic trioxide (ATO) in combination with other chemotherapeutic drugs, such as etoposide and paclitaxel, tumor growth seems to be controlled in 75% of cases [82].

Gli1 can also be pharmacologically inhibited by two compounds, GANT58 and NCS75503, which act as Gli1 antagonists [83].

### 6.4. Mammalian Target of Rapamycin (mTOR)

The mammalian target of rapamycin (mTOR) is a potential target in ES because it was highlighted that ES expresses and upregulates, by phosphorylation, this crucial pathway for cancer cell proliferation and survival [84].

Some mTOR inhibitors (mTORi), such as rapamycin, deforolimus, and MLN0128, have shown modest activity when they were used as monotherapy, most likely because the inhibition of mTOR leads to the activation of AKT through the negative feedback inhibition mediated in part by IGF1R. It is possible that overcoming this resistance mechanism, with the combination of mTORi and IGF1Ri, could guarantee a higher anticancer activity [85].

## 7. Homeostasis and Metabolism in ES

ES cells, in order to survive, must maintain a balance between the production and degradation of the EWSR1-FLI1 protein. Since the half-life of this protein is 4 h, ES cells must produce it continuously to keep pace with its degradation.

Protein degradation is performed through the proteasome system and ubiquitination mechanisms.

To deplete ES cells from EWSR1-FLI1 proteins and consequently reduce their accumulation and misfolding, two strategies may be applied: facilitating ubiquitination or blocking the de-ubiquitination. Ubiquitination serves as a biological mark to guide misfolded and accumulated proteins through proteasomal degradation [86].

### 7.1. Heat-Shock Protein 90 (HSP90)

The modulation of EWSR1-FLI1 protein expression may be reached with the inactivation of chaperones, small molecules that participate in protein homeostasis in normal cells. Among chaperones, Heat-shock protein 90 (HSP90) plays a crucial role in the response to protein misfolding and aggregation. When EWSR1-FLI1 proteins are produced, HSP90 works to guarantee the pre-mentioned homeostasis.

Zelavespib, a HSP90 inhibitor, disrupts this equilibrium and therefore there is an increase in EWSR1-FLI1 turnover and a decrease in ES cell viability, a mechanism that was preclinically proved in in vitro studies both alone and in combination with bortezomib [87].

The aim of this study was to investigate the activity of PU-H71, named zelavespib, in multiple patient-derived ES cell lines.

The drug demonstrates the inhibition of ES proliferation via activation of pro-apoptotic mechanisms.

HSP90 needs a chemical reaction in order to be activated; this reaction involves acetylation.

Taking into consideration what was previously said, entinostat or other histone deacetylase inhibitors, not only could be useful as epigenetic modulators, as previously mentioned, but also as a target of the protein homeostasis system [88] (Figure 3B).

### 7.2. Proteolysis Targeting Chimeric Molecules (PROTACs)

Degradation of EWSR1-FLI1 proteins may be also achieved through the interaction with Proteolysis Targeting Chimeric Molecules (PROTACs), even if no compound of this class has been tested in clinical trials yet.

PROTACs are bi-functional compounds comprising two parts: one portion has the role of binding specific target proteins while the other acts as a chaperone, conducting the targets to poly-ubiquitination and degradation [89].

Preclinical evidence shows promising activity of BETd, a PROTAC directed against BRD4, a transcriptional and epigenetic regulator essential during ES development [90].

### 7.3. Unfolded Protein Response and IRE1α-XBP1 Inhibitors

Biologically speaking, the synthesis, the correct folding and maturation of proteins, occurs in the endoplasmic reticulum (ER).

When misfolded or unfolded proteins are produced, they induce the so-called ER stress that enhances the activation of the unfolded protein response (UPR). In the signaling pathway of UPR, inositol-requiring enzyme 1 (IRE1α) plays a major role as it is a sensor protein which serves as a kinase and endoribonuclease located in the ER membrane [91].

During ER stress, IRE1α undergoes oligomerization and when tetramers are formed, the X-box binding protein 1 (XBP1) splicing reaction occurs.

In fact, upon activation of IRE1α, its ribonuclease (RNase) domain triggers the unconventional splicing of XBP1 (XBP1s) mRNA and, therefore, the homeostatic transcription factor XBP1s is produced. This mechanism promotes cell survival via upregulation of pro-survival pathways in cancer cells [92].

Proteomic studies of ES have showed that XBP1, a protein playing a critical role in the UPR in the ER, is highly expressed in surgical samples of ES.

IRE1α-XBP1 inhibitors, such as toyocamycin, 2-hydroxy-1-naphthaldehyde (HNA), STF-083010 (STF), and 3-ethoxy-5 6-dilbromosalicylaldehyde (3ETH), seem to suppress cell viability both in vitro and in vivo (Figure 3C).

As a matter of fact, IRE1α-XBP1 inhibitors may be useful therapeutic options for ES patients [93].

### 7.4. Warburg Effect Inhibition: Lactate Dehydrogenase Inhibitors (LDHi)

The Warburg effect is an established hallmark of cancer. Cancer cells are dependent on anaerobic glycolysis in order to exploit their functions, but this dependency is shared with other cells, like mature erythrocytes. Anaerobic glycolysis needs specific enzymes, and, among them, lactate dehydrogenase (LDH) plays a critical role.

ES is not an exception in these terms and, as a matter of fact, LDH is a negative prognostic factor in ES patients and a promising target [94].

LDH inhibitors have been demonstrated to induce necrosis in ES models but their implementation in clinical practice is limited by hemolysis, a universal dose-limiting side effect of this class of drugs [95].

## 8. Immunotherapy in ES

The immune system elicits its role against cancerous cells through different mechanisms. Unfortunately, at some point cancer cells succeeded in escaping from immune surveillance. This evading capacity can be both innate and acquired even if it is more frequent that, under selective pressure, malignant cells gain features that enhance their escape from the immune system.

Immunotherapy in oncology has the aim of establishing a major response of the immune system of the host against cancer cells.

To target cancer cells, T-cells, the main effector of the adaptive immune response, need antigen presentation by the major histocompatibility complex (MHC), expressed on qualified antigen presenting cells (APCs).

ES cells derived from advanced patients exhibit a defective antigen presentation and fail to express MHC, resulting in immune tolerance [96].

Furthermore, T-cells, both CD4+ and CD8+, taken from ES patients showed an exhausted phenotype when compared to healthy donors, confirmed by a more pronounced programmed cell death 1 (PD-1) expression [97]. This exhausted phenotype could be also explained by the fact that corrective apoptosis pathways, such as the TNF-related apoptosis inducing ligand (TRAIL) pathway and the death receptor pathway are still active and more pronounced in these cells [97].

Another reason that may explain the immune tolerance of ES cells is the existence, in the tumor microenvironment, of myeloid-derived suppressor cells (MDSCs) and immunosuppressive fibrocytes which elicited their activity through several mechanisms including nutrient depletion, oxidative stress, activation of regulatory T-cells (T-reg), and the inhibition of chimeric antigen receptor T-cells (CART) [97].

ES cells are physiologically sensitive to the immunosuppressive action of natural killer (NK) and cytotoxic T-cell (CD8). Consequently, tumors showing a higher number of cytotoxic T-cells have a better prognosis in terms of OS [98]. Moreover, NK cells do not rely on antigen presentation to be effective, thus they bypass the mentioned mechanisms of immune tolerance.

These assumptions represent a strong biological rationale for the implementation of immunotherapy in ES, particularly for patients that do not seem to respond to conventional chemotherapeutic drugs.

Immunotherapies for ES include, but are not limited to mAbs, immune cell-based therapies, and cancer vaccines (Cv).

### 8.1. Immune Checkpoint Inhibitors (ICIs)

Immune checkpoint inhibitors (ICIs), such as pembrolizumab, a mAb directed against PD-1, have failed in demonstrating a significant clinical activity in adults with ES. The reasons why this agent did not confirm the same exciting results as in other solid tumors could be due to various factors: the low mutational burden, the lack of PD-L1 expression, the HLA loss, and the absence of reactive T-cell in tumor microenvironment that characterize ES cells.

Pembrolizumab failure stems from SARC028, the first prospective open-label phase II multicenter study where none of the thirteen enrolled ES patients showed any clinical benefit from anti-PD-1 blockade [99]. This failure in the ES cohort could be related to the highly suppressive immune microenvironment in the tumor as previously underlined. The fact that only one patient with ES had demonstrated stable disease (SD) suggests that immune biomarkers could help to select patients that may benefit, at least in part, from ICI single agent treatment. Furthermore, poor activity in the monotherapy of ICIs could be increased with combination therapies as discussed below.

In a phase I/II trial nivolumab, another PD-1 inhibitor, was investigated as monotherapy in terms of safety and anti-tumor activity in children and young adults with recurrent or refractory tumors including ES [100].

As for SARC028, ICI monotherapy with nivolumab did not show significant anti-tumor activity although it was well tolerated in terms of toxicities.

In order to increase the efficacy of ICIs in sarcomas, the immunological and immunomodulatory properties of conventional CT and target therapies (TT) have been investigated. Metronomic cyclophosphamide has been described as being able to restore T-cells and NK effectors while inhibiting T-regs [101].

Although apparently promising, this strategy has not led to important results in terms of OR and PFS rates [102]. In a single arm phase II trial axitinib, a selective TKI, was combined with pembrolizumab in patients with advanced or metastatic sarcoma. Most patients with partial response (PR) belonged to the alveolar soft part sarcoma (ASPS) group.

The benefit in the non-ASPS group, in terms of PFS and OR, was limited, raising the question of whether the addition of an ICIs has any value in non-ASPS histology [103]. It is worth noting that only one patient in the trial had a diagnosis of ES.

### 8.2. Tumor-Reactive T-Cells

In recent unsuccessful trials, tumor-reactive T-cells have been used but there is a main limit: ES cells downregulate MHC-1 on their surfaces reducing the recognition of intracellular antigen from T-cells as a defense mechanism to avoid the immune response.

MHC-1 expression may be induced by IL-12 or INF-γ, two cytokines that could play a fundamental role in future trials dealing with immunotherapy in ES [104]. Tumor-reactive T-cells can be obtained in two ways: one requires the selection of the autologous adoptive T-cell which recognizes the tumor followed by its expansion in vitro and eventually its transfer into the patient. Although intriguing, this process could be challenging, time consuming, and possibly inefficient. The other way consists in isolating TCRs from T-cells with proven tumor reactivity and using them to transduce polyclonal autologous T-cells, transferred after expansion into patients whose tumors express the given antigens. Several pre-clinical trials have tried to translate these approaches into practice with disappointing results [105,106,107]. It is possible that the identification of combination therapy and the implementation of strategies with the aim of rescuing HLA downregulation could increase ES cells sensitivity to these strategies.

### 8.3. Chimeric Antigen Receptor T-Cell (CAR-T)

In order to overcome these limits, the chimeric antigen receptor T-cell (CAR-T) has been used because of its independence from MHC-1 recognition. CAR-T cells are T-cells engineered to express CARs on their surfaces. CARs are made up by an antigen binding domain and a T-cell activation domain.

CAR-T cells have been developed against the vascular endothelial growth factor receptor 2 (VEGFR2), IGF1R [108], ganglioside2 (GD2) [109], hepatocellular receptor tyrosine kinase class A2 (EphA2) [110]; all these antigens have presented as highly expressed on ES cells’ surface. Preclinical studies targeting GD2 demonstrated a reduction in tumor volume and clinical trials with the more promising CAR-T cells have been initiated [111]. (Figure 4A,B)

### 8.4. Cancer Vaccines

Vaccines have been also explored as immunotherapeutic strategies. Cancer vaccines consist of peptides, proteins, or lysates of tumor cells, sometimes mixed with certain adjuvants to increase their immunogenicity. Their role of attracting dendritic cells (DCs) and CCR7-positive T-cells in the site of the tumor has the aim of converting the immune environment of ES from a “cold” one to an “hot” one. This process eliminates the suppressive immune activity of ES and enhances immune response. This happens as an example throughout the inhibition of immune inhibitory proteins such as transforming growth factor β1 (TGF-β1) [96].

DCs, pulsed with peptides derived from the EWSR1–FLI1 fusion protein and adjuvant intravenous compounds that stimulate the immune system such as IL-2, were used in clinical trials but tumor regression was time-limited [112].

In a phase I trial tumor lysate pulsed DCs were investigated in children with relapsed solid malignancies [113]. Among them, two had a diagnosis of ES and showed progressive disease (PD) during treatment.

Another vaccination strategy is represented by the FANG (or Vigil) immunotherapy. This complex option relies on tumor antigens release, DCs recruitment, and the activation and enhancement of tumor cells migration to local lymph nodes and many more effects [114].

After being evaluated in a phase I trial Vigil has demonstrated to be well tolerated, to elicit a tumor-specific systemic immune response, and to be associated with an objective response [115]. Indeed, when compared with ES patients treated with conventional therapy, patients administered with Vigil showed 1-year survival of 73% compared with 23%. To date, it is under investigation in a phase III randomized trial of intradermal injection in combination with irinotecan and temozolomide versus combination chemotherapy alone in patients with metastatic ES, refractory/intolerant or recurrent to one prior line of chemotherapy (NCT03495921). Table 1 lists other potential new targets in the treatment of ES that are underway or have yet to undergo pre-clinical and/or clinical investigation for advanced and refractory disease. (Trialgov.it).

## 9. Discussion

ES is a complex disease. Clinical experiences with ES of the thoracic cavity have only be reported as small case se-ries and case reports (cit) highlighting the importance of a deeper knowledge of this rare entity in medical oncology [116]. Despite the multidisciplinary strategies involving both local therapies, such as surgery and radiotherapy, and systemic therapy in extending OS and disease-free survival, particularly in early stages, ES tends to recur and no valid options have demonstrated to be effective.

The extreme rarity of this disease has limited the design of tailored trials aimed to shedding a light on the best therapeutic strategies at recurrence.

The complex and heterogenous molecular biology that permeates ES represents a prolific source for promising, but still embryonic, therapeutic strategies emphasizing the need to move from a traditional molecular-agnostic approach to a molecular-driven strategy.

Fusion transcript of the chimeric gene EWSR1-ETS is considered the main molecular alteration in ES and its downstream effects should be deeply understood to investigate tailored strategies and original compounds that may affect one or more specific pathways active in this disease.

The inhibition of RTKs, both with mAbs [17,18,19,20,22,23,30,34,35] and small molecules [30,31,32,34], have shown preliminary encouraging results especially when combined with conventional CT [27,28,29] or other TT [26].

Unfortunately, heterogeneity in receptor expression, the dependency of ES cells on different receptors, the existence of escape and resistance mechanisms, and the peculiar toxicity profiles of these agents still represent some of the reasons why these drugs have not reached clinical practice yet.

EWSR1-ETS expression has been investigated as a potential target. Antisense oligodeoxynucleotides [41] (oligos) and small interfering RNA (siRNA) [44] interfering with the mRNA that codes for the fusion protein have been investigated even if pharmacokinetic and pharmacodynamic concerns remain to be solved.

To date, small molecules, YK-4-279 and TK216, that disrupt the interaction between EWSR1-ETS and other transcriptional modulators are in more advanced phases of development [46,47]. Their implementation in clinical practice is prevented by the fact that it is not clear whether EWSR1-ETS translocation is the only genetic alteration sensitive to these agents. Unknown resistance mechanisms could also reduce their efficacy, suggesting a possible role for combination therapies.

Nowadays it is clear that ES is not solely driven by genetic alteration but also by epigenetic change.

The understanding of the ES epigenetic landscape has led to the investigation of compounds directed against proteins and/or complexes involved in chromatin remodeling mechanisms such as LSD1 inhibitors [52], HDM inhibitors [53], HDAC inhibitors [58], and BET inhibitors [59]. Some concerns about their hematologic and non-hematologic toxicities as well as the existence of resistance mechanisms have prevented their clinical implementation deserving further investigations.

The proliferative rate of ES cells represents a source of endogenous stress and DNA damage that need the integrity of the DNA damage response systems. With this rationale, agents that target one or more of these repair systems, such as PARPi [66,67,68,69] and ATRi [72,73], are being investigated both alone and in combination with CT in several clinical trials. Preliminary results suggest that the knowledge and availability of predictive markers could help to better select patients who could benefit from these compounds.

Replication is not the only stressful activity in ES. Because of the unexhausted production of the EWSR1-ETS chimeric protein, there is an activation of the ER that responds to the accumulation of misfolded proteins. The UPR system can be inhibited by both targeting chaperones [87] that guide the correct folding of the proteins, the proteasome [87] and IRE-XPB1 pathways [91,92,93].

In order to survive and proliferate ES cells rely on anaerobic glycolysis and on the activity of LDH, an enzyme involved in this metabolic process. LDH inhibitors [95] have been studied even if hemolysis represents an important limitation for their use in clinical practice.

Moreover, the complex relation between the immune system and ES has been studied. Although no enthusiastic results have been shown with ICIs in monotherapy [99,100], combination therapy or other strategies involving CAR-T [111] and vaccines [112,113] are under pre-clinical and clinical examination in ES.

As mentioned before, the poor knowledge of predictive biomarkers, able to select patients that may benefit from these strategies, has prevented obtaining the awaited results.

## 10. Conclusions

ES is a complex disease, and its treatment requires dedicated knowledge and the highest expertise. ES lacks dedicated trials designed in order to address unmet oncology needs and the biology that underlies and permeates its clinical behavior is widely still unknown and not fully understood.

Multidisciplinary approaches, involving both local treatments and systemic therapies, especially when carried out in dedicated centers, have improved survival rates and other clinical outcomes especially when disease is detected in early stages.

Unfortunately, ES tends to recur and when this happens no valid options are currently available.

Deepening the knowledge of ES biology could surely represent a key to improving clinical outcomes in patients with refractory or recurrent disease after the failure of conventional therapies.

In order to design new compounds capable of changing the natural history of this neoplasm it is mandatory to deeply understand the critical intracellular pathways on which ES proliferation relies, besides its epigenomic profile and editing and the complex relation between the tumor, the immune system and microenvironment.

## Figures and Tables

**Figure 1 biomedicines-11-00874-f001:**
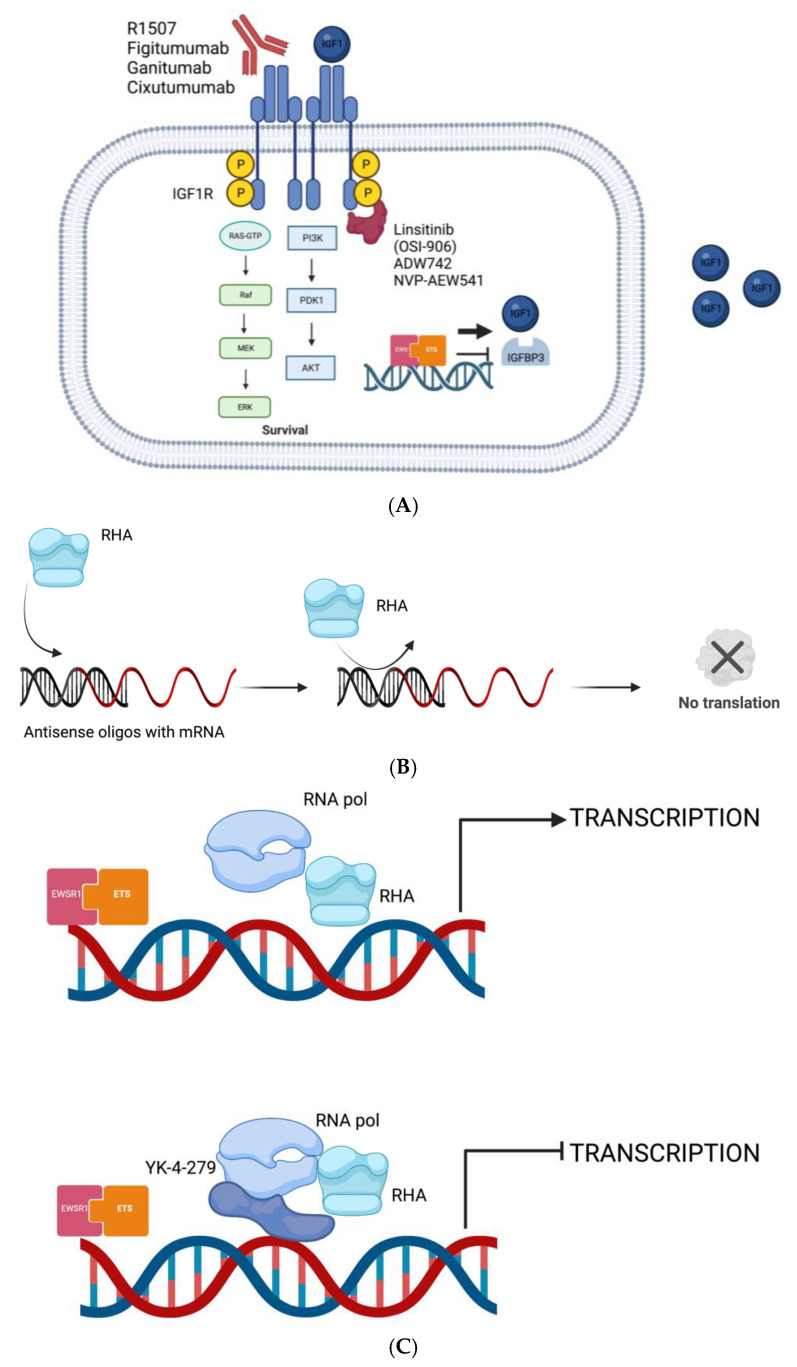
(**A**) EWSR1-ETS transcription activity enhances the transcription of IGF1 and reduces the transcription of IGFBP3; IGF1 binds to IGF1R and activates MAPK pathway leading to proliferation and survival; antibodies and small molecules that inhibit IGF1R signaling. (**B**) The mRNA, derived from the transcription of the fusion gene, is translated into the oncogenic fusion protein EWSR1-ETS that elicits its downstream effect; the mRNA translation process is stopped by oligos, and siRNAs and the fusion protein is no longer produced. (**C**) EWSR1-ETS depends on RHA to enhance its transcriptional activity; YK-4-279 can alter the interaction between EWSR1-ETS and RNAH resulting in transcription stop. (Created with BioRender.com).

**Figure 2 biomedicines-11-00874-f002:**
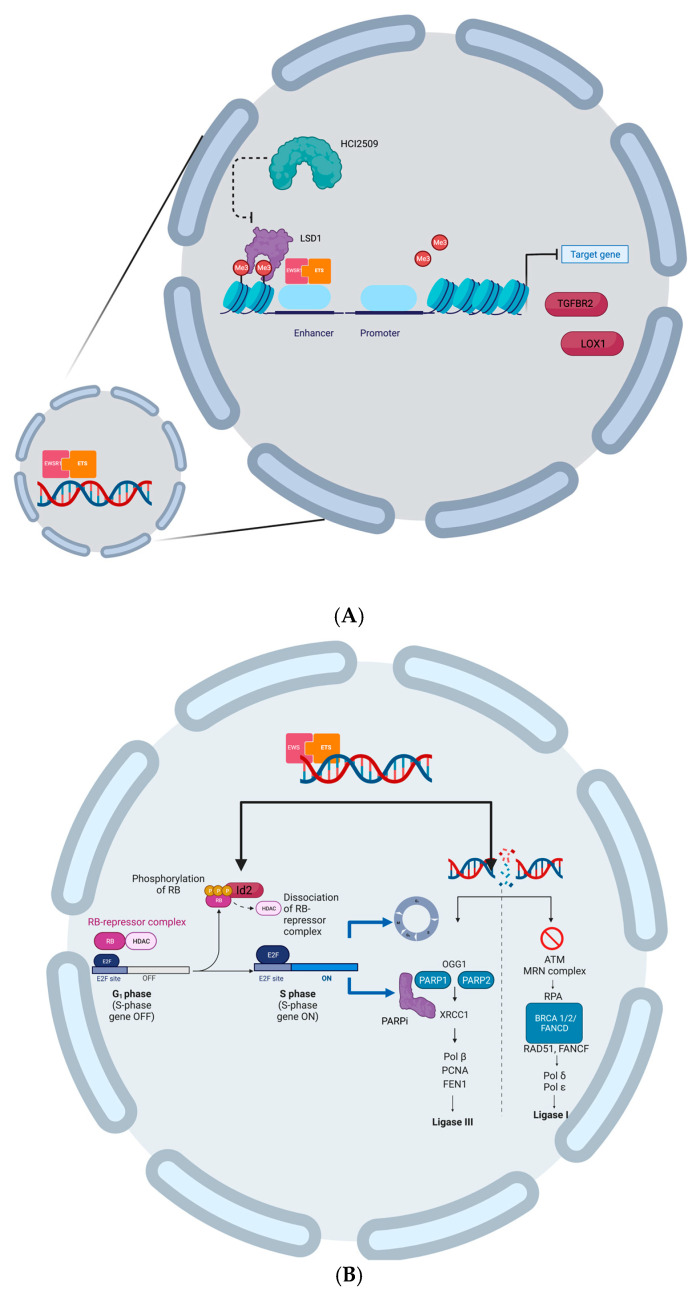
(**A**) EWSR1-ETS uses LDS1 to repress the expression of two target genes LOX and TGFBR2 that act as tumor suppressors; HCL2509 can inhibit LSD1 activity and reverse transcriptional activity of EWSR1-ETS. (**B**) EWSR1-ETS induces Id2 that releases E2f from RB inhibition; E2f induces cell cycle progression and PARP1 expression; replication and transcriptional stress lead to DNA damage that cannot be repaired either by PARP1, targeted by PARPi, nor by BRCA1 that is functionally absent in ES cells. (Created with BioRender.com).

**Figure 3 biomedicines-11-00874-f003:**
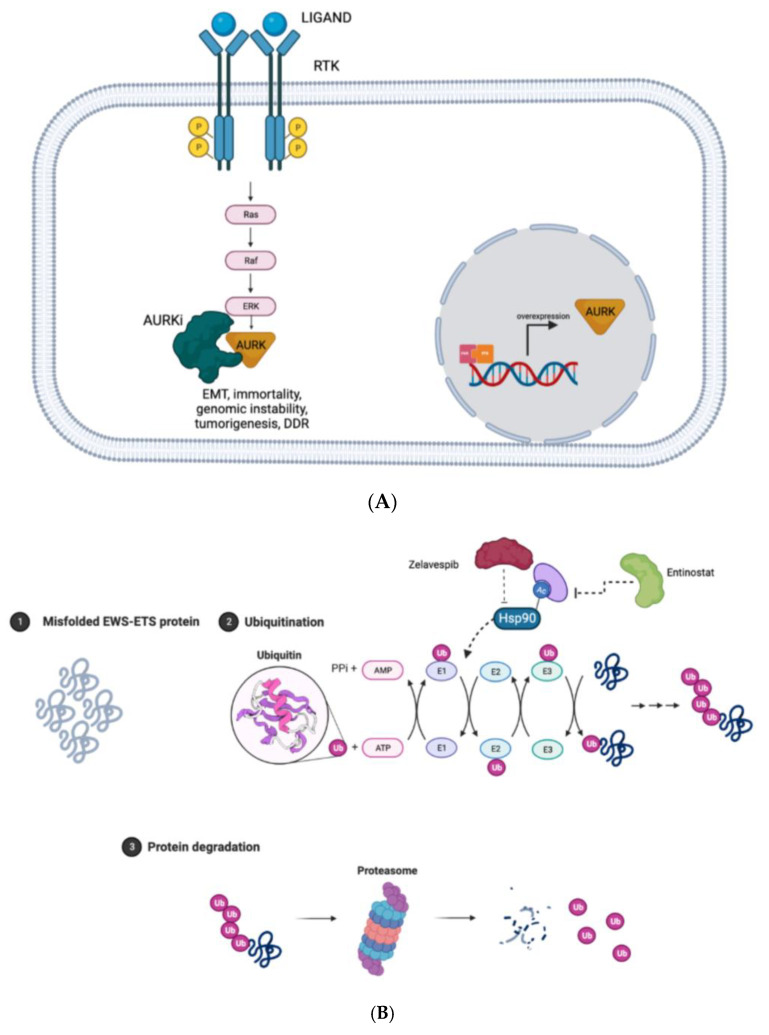
(**A**) Mechanisms of activation and inhibition of AURKs and its downstream effects. (**B**) Misfolded and accumulated EWSR1-ETS protein are ubiquitinated with the help of chaperones protein and directed to proteasome system in order to be degraded and guarantee homeostasis in proliferating cells; when Zelavespib, a HSP90i, and Entinostat, a HDACi, inhibit the ubiquitination of misfolded and accumulated proteins via proteasome, cells died due to proteotoxic stress. (**C**) ER stress induces oligomerization/tetramerization of IRE1 in ER leading to the unconventional splicing of XPB1 in XPB1s that enhances pro-survival signals; IRE1α-XBP1 inhibitors inhibit this process suppressing viability of cancerous cells. (Created with BioRender.com).

**Figure 4 biomedicines-11-00874-f004:**
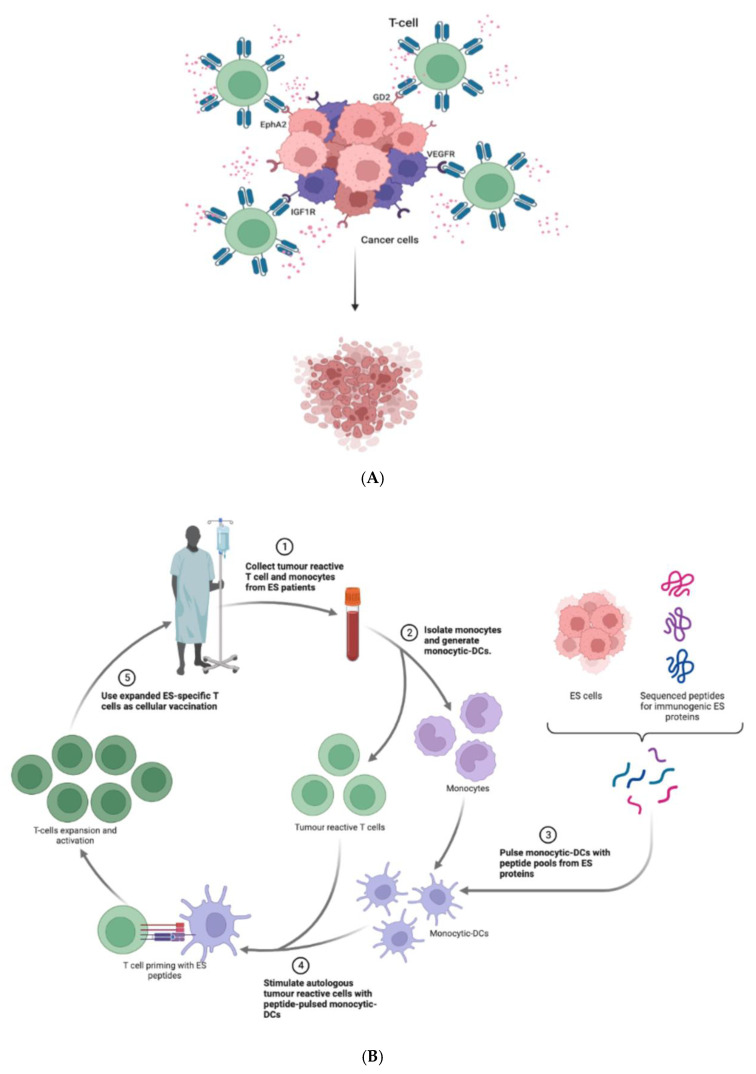
(**A**) e (**B**) Immunotherapy in ES. (Created with BioRender.com).

**Table 1 biomedicines-11-00874-t001:** Other potential new targets in the treatment of ES that are undergoing pre-clinical and/or clinical investigation for advanced and refractory disease. (Trialgov.it).

NCT	Target	Function in ES	Therapy	Status
NCT05275426	Chk1	DNA repair	Chk1 inhibitor (LY2880070) plus gemcitabine	ongoing
NCT03416517	multi TK (VEGFR, PDGFR, C-KIT, FGFR)	Angiogenesis	Anlotinib	unknown
NCT04183062	MARCKS (myristoylated alanine-rich C-kinase substrate)	Differentiation and tumor promoting function of cancer associated fibroblast (CAFs)	BIO-11006	active, not recruiting
NCT05440786	CDK4/6	Cell proliferation	Abemaciclib	recruiting
NCT02243605	multi TK (c-Met, VEGFR2, AXL e RET)	Proliferation and cell survival	Cabozantinib	active, non recruiting
NCT04901702	PARP	DNA repair (after onivyde damage)	Onivyde + Talazoparib	recruiting
NCT05395741	VEGFRs	Angiogenesis	Regorafenib	recruiting
NCT04890093	HSP90	Misfolding protein response	PEN-866	not yet recruiting
			(contains SN-38)	
NCT05093322	VEGFR1, VEGFR2, VEGFR3, FGFR1, and CSF-1R	Cell survival and proliferation	Surufatinib	recruiting
NCT03373097	GD-2	Cell survival and proliferation	Anti-GD2 CAR T Cells	recruiting
NCT03715933	DR5	Cell death	INBRX-109 (DR5 agonist)	recruiting
NCT05159518	CDK9	Epigenetic and transcriptional reprogramming	PRT2527	recruiting
			(CDK9i)	
NCT04308330	HDAC	Epigenetic and transcriptional reprogramming	Vorinostat	recruiting
NCT05071209	ATR	DNA repair	Elimusertib	recruiting
NCT04195555	IDH1/2	Cell metabolism	Ivosidenib	recruiting
NCT04320888	RET	Cell proliferation and survival	Selpercatinib	recruiting
NCT03698994	ERK1/2	Cell proliferation and survival	Ulixertinib	active not recruiting
NCT04284774	farnesyltransferase inhibitor	Cell proliferation	Tipifarnib	recruiting
NCT04851119	TBL1 (Transducin Beta Like Protein 1)	Cell proliferation and metastasization	Tegavivint	recruiting
NCT03213665	EZH2 or the SWI/SNF complex	Epigenetic regulation	Tazemetostat	active not recruting
NCT03213652	ALK and ROS1	Proliferation and cell survival	Ensartinib	recruiting
NCT03213704	NTRK	Proliferation and cell survival	Larotrectinib/Entrectinib	recruiting
NCT04897321	B7H3	Immune response	B7H3 CAR T Cell	recruiting
NCT03220035	BRAFV600E	Proliferation and cell survival	Vemurafenib	active not recruiting
NCT03618381	EGFR	Proliferation and cell survival	EGFR806 CAR T	recruiting
NCT03213678	TSC or PI3K/mTOR	Proliferation and cell survival	Samotolisib	recruiting
NCT03425279	AXL (RTK)	Proliferation and cell survival	BA3011 (CAB-AXL-ADC)	recruiting
NCT05182164	TK and immune environment	Proliferation and cell survival and immunotherapy	Pembrolizumab + cabozantinib	recruiting

## Data Availability

Not applicable.

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
