# Peer review of "Shifting from a Biological-Agnostic Approach to a Molecular-Driven Strategy in Rare Cancers: Ewing Sarcoma Archetype"

_biomedicines, 2023, doi:10.3390/biomedicines11030874_

Round 1

Reviewer 1 Report

This review focuses on Ewing Sarcoma, a very rare bone cancer that remains challenging to cure. It is important to acknowledge that several articles have already been published on similar topics (PMCID: PMC9032664, PMID: 30349420). While this manuscript tries to emphasize on translational aspects by providing updates on clinical trials, the current format is very limited and poorly organized.

1- The title is highly similar to a published review (PMID: 35454895).

2- The entire manuscript is poorly organized. For instance, the authors describe some ongoing clinical trials in the introduction before dedicating a section on emerging treatment strategies. There, they mix targets (e.g. RTKs), mechanisms of resistance, clinical trials, antibody-based inhibitors, and small molecule inhibitors, etc.

3- For all figures: the summary graphs are difficult to read due to their very small sizes.

4- The summary table is similar to the one published in PMID: 35454895. Recent trials should have been highlighted to emphasize on newer clinical directions.

5- Each sentence is presented as a small paragraph. 

6- The section about immunotherapy is interesting but not well organized.

Reviewer 2 Report

This is a descriptive review article describing the development of a novel therapy based on molecular biological features with respect to Ewing sarcoma. Although I feel that it is relatively well written, I believe that the following points need to be revised.

#1. The descriptions of thoracic cavity (Line 17) and thoracic sarcoma (Line 19) in the Abstract have the risk of giving the reader a false impression that the article is about Ewing's sarcoma of the thoracic cavity. As you know, Ewing sarcoma is not a tumor that occurs exclusively in the thoracic cavity, but rather a tumor that occurs much more frequently outside of the thoracic cavity. I ask for your review and correction.

#2. In the Introduction, it is stated that the diagnosis of Ewing's sarcoma is based on immunohistochemistry (Lines 41-45), which is not in line with the current situation. It is now common knowledge that the detection of fusion genes is essential for an accurate diagnosis of Ewing sarcoma.

#3. Lines 70-71 state that there is no regimen that improves the PFS of Ewing sarcoma, but this is not true. For example, the rEECur trial also reported prolonged PFS. The description needs to be reviewed and revised.

#4. Lines 227, 242, etc. refer to EWS-FLI1, but the terminology should be consistent and the correct term should be EWSR1-FLI1.

#5. Lines 227-241 mention the reduction of EWSR1-FLI1 expression, but there is no mention of how the reduction of EWSR1-FLI1 expression affects the proliferative and metastatic potential of Ewing sarcoma cells. This point needs to be mentioned.

#6. In Line 367, the abbreviation "ES" is given to epithelioid sarcoma, which is the same as the abbreviation "ES" for Ewing sarcoma. This is undesirable for a scientific paper.

#7. SARCS028 in Lines 649 and 660 should be SARC028 correctly.

#8. In Line 649, parentheses are missing in reference #100.

#9. The text in Figures is extremely small and difficult to read. Please consider rearranging or re-creating the figures.

Round 2

Reviewer 1 Report

The revised version is better organized and contains up-to-date studies relevant to ES.